# DAO-CP: Data-Adaptive Online CP decomposition for tensor stream

**Sangjun Son**[ID][☯], **Yong-chan Park**[☯], **Minyong Cho, U. Kang**[ID]*

Seoul National University, Seoul, Republic of Korea

☯ These authors contributed equally to this work.
* ukang@snu.ac.kr

**Data Availability Statement:** The data underlying the results presented in the study are available from https://github.com/snudatalab/DAO-CP.

**Funding:** This work was partly supported by the National Research Foundation of Korea(NRF) funded by MSIT(2022R1A2C3007921) and MSIT

## Abstract

How can we accurately and efficiently decompose a tensor stream? Tensor decomposition is a crucial task in a wide range of applications and plays a significant role in latent feature extraction and estimation of unobserved entries of data. The problem of efficiently decomposing tensor streams has been of great interest because many real-world data dynamically change over time. However, existing methods for dynamic tensor decomposition sacrifice the accuracy too much, which limits their usages in practice. Moreover, the accuracy loss becomes even more serious when the tensor stream has an inconsistent temporal pattern since the current methods cannot adapt quickly to a sudden change in data. In this paper, we propose DAO-CP, an accurate and efficient online CP decomposition method which adapts to data changes. DAO-CP tracks local error norms of the tensor streams, detecting a change point of the error norms. It then chooses the best strategy depending on the degree of changes to balance the trade-off between speed and accuracy. Specifically, DAO-CP decides whether to (1) reuse the previous factor matrices for the fast running time or (2) discard them and restart the decomposition to increase the accuracy. Experimental results show that DAO-CP achieves the state-of-the-art accuracy without noticeable loss of speed compared to existing methods.

## Introduction

*Given a tensor stream, how can we decompose it accurately and efficiently?* A multi-dimensional array or *tensor* has been a fundamental component for numerous applications including signal processing [1–3], computer vision [4–6], graph analysis [7, 8], and statistics [9]. Tensor decomposition is a generalization of the matrix decomposition, and plays an important role in latent feature discovery and estimation of unobservable entries [10–12].

A tensor is called *dynamic* if its size and value change over time (for example, time-evolving network traffic data, social networks, and so on), or *static* otherwise [13]. Analysis of dynamic tensors with tensor decomposition is a crucial task. However, the methods devised for static tensor decomposition cannot be easily applied for dynamic tensor analysis since such static methods perform many iterations of computation until convergence at every time step, which leads to prohibitively large time cost.

(2019R1A2C2004990). This work was partly supported by Institute of Information & communications Technology Planning & Evaluation (IITP) grant funded by the Korea government(MSIT) [NO.2021-0-02068, Artificial Intelligence Innovation Hub (Artificial Intelligence Institute, Seoul National University)] and [NO.2021-0-01343, Artificial Intelligence Graduate School Program (Seoul National University)]. The Institute of Engineering Research and ICT at Seoul National University provided research facilities for this work. The funders had no role in study design, data collection and analysis, decision to publish, or preparation of the manuscript.

**Competing interests:** The authors have declared that no competing interests exist.

On the other hand, dynamic tensor decomposition methods aim to incrementally and quickly analyze tensors; however, the accuracy of existing methods is not satisfactory. Indeed, current dynamic methods (1) bypass the factor update for temporal mode to improve the speed [14], or (2) decompose the current tensor slice using prior factor matrices [15, 16]. Unfortunately, these methods suffer from poor accuracy when the tensor stream has an inconsistent temporal pattern because they cannot adapt quickly to abrupt changes in data [17, 18].

In this paper, we propose Data-Adaptive Online CP decomposition (DAO-CP), an accurate and efficient tensor stream decomposition algorithm which adapts to data changes. The main ideas of DAO-CP are to (1) detect change points of "themes" in a tensor stream by tracking local error norms, and (2) re-decompose the tensor stream whenever a new theme is discovered. DAO-CP automatically decides whether to reuse the previous results of decomposition or to discard them depending on how much changes are detected in the tensor stream. Consequently, it provides much more accurate and fast decomposition for real-world datasets even with inconsistent temporal patterns. Furthermore, we introduce complementary matrices in order to reduce the redundant computations in CP-ALS optimization. We also simplify the estimation loss function from DTD [15] by fixing the non-temporal modes. As a result, DAO-CP achieves lower time complexity than the existing methods without accuracy loss. Through experiments, we show that DAO-CP outperforms the current state-of-the-art algorithms in terms of accuracy with little sacrifice in running time. We also investigate the sensitivity and the effect of hyperparameters of our proposed method.

The main contributions are summarized as follows:

- **Method**. We propose DAO-CP, an accurate and efficient online method for tensor stream decomposition.

- **Analysis**. We theoretically analyze the computational complexity of DAO-CP and compare it to existing methods.

- **Experiments**. DAO-CP shows the state-of-the-art accuracy on both synthetic and real-world datasets without significant loss of speed (see Fig 1).

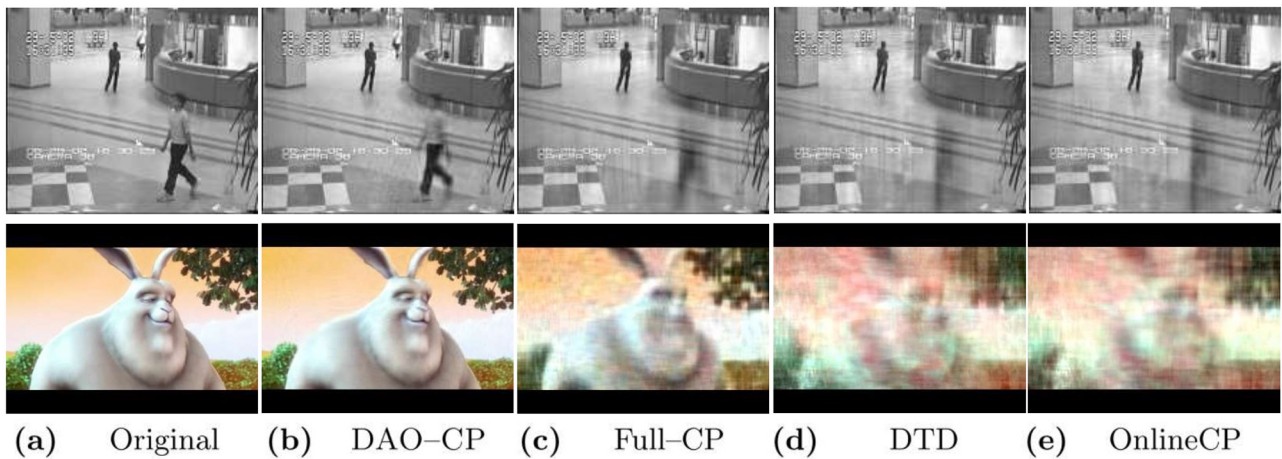

**Fig 1. Accuracy comparison between DAO-CP (proposed) and its competitors on Airport Hall (upper) and Sample Video (lower) datasets.** DAO-CP automatically detects a change of theme (for example, an object starts moving or a scene changes) and re-decomposes the data depending on the degree of changes. Note that DAO-CP results in much more clear images than the competitors with little sacrifice in speed.

**Table 1. Table of symbols.**

| Notation | Definition |
|---|---|
| $\mathbf{A}$ | matrix |
| $\mathbf{A}^{\top}$ | transpose of $\mathbf{A}$ |
| $\mathbf{A}^{\dagger}$ | pseudoinverse of $\mathbf{A}$ |
| $I_n$ | length of the $n$-th mode of tensor |
| $R$ | rank of tensor |
| $\boldsymbol{\mathcal{X}} \in \mathbb{R}^{I_1 \times \cdots \times I_N}$ | $N$-th order tensor |
| $\mathbf{A}_n \in \mathbb{R}^{I_n \times R}$ | factor matrix for $n$-th mode of tensor |
| $\|\boldsymbol{\mathcal{X}}\|$ | Frobenius norm of $\boldsymbol{\mathcal{X}}$ |
| $\mathbf{X}_{(n)} \in \mathbb{R}^{I_n \times \prod_{i \neq n} I_i}$ | mode-$n$ unfolding matrix of $\boldsymbol{\mathcal{X}}$ |
| $[\![\cdot]\!]$ | Kruskal operator, e.g. $\boldsymbol{\mathcal{X}} \approx [\![\mathbf{A}_1, \cdots, \mathbf{A}_N]\!]$ |
| $\otimes$ | Kronecker product |
| $\odot$ | Khatri-Rao product |
| $\circledast$ | element-wise product (Hadamard product) |
| $\oslash$ | element-wise division |

The code and datasets are available at https://github.com/snudatalab/DAO-CP. The rest of this paper is organized as follows. We first demonstrate preliminaries of tensor decomposition algorithms. We then present our proposed method in detail. After showing experimental results, we discuss related works, and conclude the paper.

## Preliminaries

We describe preliminaries of tensors and tensor decomposition algorithms. Table 1 summarizes the symbols used in this paper.

### Tensors

Tensors are multi-dimensional arrays that generalize vectors (1-order tensors) and matrices (2-order tensors) to higher orders. We denote vectors with bold lowercase letters ($\mathbf{a}$), matrices with bold capital letters ($\mathbf{A}$), and tensors with bold calligraphic letters ($\boldsymbol{\mathcal{X}}$). An $N$-th order tensor $\boldsymbol{\mathcal{X}}$ has $N$ modes whose lengths are $I_1, \cdots, I_N$, respectively. A tensor can be *unfolded* or *matricized* along any of its modes [19], and the unfolded matrix of $\boldsymbol{\mathcal{X}}$ along the $n$-th mode is denoted by $\mathbf{X}_{(n)}$. When a tensor is unfolded, its elements are reordered into a matrix form; the mode-$n$ unfolding matrix $\mathbf{X}_{(n)} \in \mathbb{R}^{I_n \times \prod_{i \neq n} I_i}$ of a tensor $\boldsymbol{\mathcal{X}} \in \mathbb{R}^{I_1 \times \cdots \times I_N}$ maps the $(i_1, \cdots, i_N)$-th element of $\boldsymbol{\mathcal{X}}$ to the $(i_n, j)$-th element of $\mathbf{X}_{(n)}$, where

$$j = 1 + \sum_{1 \leq k \leq N, \; k \neq n} \left[ (i_k - 1) \prod_{1 \leq m \leq k-1, \; m \neq n} I_m \right]. \tag{1}$$

We define the *Frobenius norm* of a tensor using the notation $\|\cdot\|$ as follows:

$$\|\boldsymbol{\mathcal{X}}\| = \sqrt{\sum_{1 \leq i_k \leq I_k, \forall k=1,\cdots,N} \left(\boldsymbol{\mathcal{X}}_{(i_1, \cdots, i_N)}\right)^2}. \tag{2}$$

In what follows, we briefly define several important matrix products. The *Kronecker product* $\mathbf{A} \otimes \mathbf{B}$ of matrices $\mathbf{A} \in \mathbb{R}^{I \times J}$ and $\mathbf{B} \in \mathbb{R}^{K \times L}$ is a matrix of size $IK \times JL$ and defined as follows:

$$
\mathbf{A} \otimes \mathbf{B} = \begin{bmatrix} a_{11}\mathbf{B} & a_{12}\mathbf{B} & \cdots & a_{1J}\mathbf{B} \\ a_{21}\mathbf{B} & a_{22}\mathbf{B} & \cdots & a_{2J}\mathbf{B} \\ \vdots & \vdots & \ddots & \vdots \\ a_{I1}\mathbf{B} & a_{I2}\mathbf{B} & \cdots & a_{IJ}\mathbf{B} \end{bmatrix}.
$$

The *Hadamard product* $\mathbf{A} \circledast \mathbf{B}$ and *Khatri-Rao product* $\mathbf{A} \odot \mathbf{B}$ are two essential matrix products used in tensor decomposition. The Hadamard product is simply the element-wise product of two matrices $\mathbf{A}$ and $\mathbf{B}$ of the same size. The Khatri-Rao product is a column-wise Kronecker product:

$$
\mathbf{A} \odot \mathbf{B} = [\mathbf{a}_1 \otimes \mathbf{b}_1, \cdots, \mathbf{a}_J \otimes \mathbf{b}_J] \in \mathbb{R}^{I_A I_B \times J}, \tag{3}
$$

where $\{\mathbf{a}_n\}$ and $\{\mathbf{b}_n\}$ denote the column vectors of $\mathbf{A} \in \mathbb{R}^{I_A \times J}$ and $\mathbf{B} \in \mathbb{R}^{I_B \times J}$, respectively.

## Tensor decomposition

CANDECOMP/PARAFAC (CP) decomposition is one of the most widely used methods for tensor decomposition, which is considered to be a key building block in many other variants [1, 20]. CP decomposition factorizes a tensor into a sum of rank-one tensors:

$$
\sum_{r=1}^{R} \mathbf{a}_1^{(r)} \circ \cdots \circ \mathbf{a}_N^{(r)}, \tag{4}
$$

where the number $R$ of rank-one tensor sets is called the *rank* of the resulting tensor. The *factor matrices* $\{\mathbf{A}_1, \cdots, \mathbf{A}_N\}$ refer to the combination of the vectors from the rank-one components, i.e.,

$$
\{ [\mathbf{a}_1^{(1)}, \cdots, \mathbf{a}_1^{(R)}], \cdots, [\mathbf{a}_N^{(1)}, \cdots, \mathbf{a}_N^{(R)}] \}. \tag{5}
$$

We express the CP decomposition result of a tensor $\mathcal{X}$ using Kruskal operator $[\![ \cdot ]\!]$ and the unfolding matrix, where the Kruskal operator provides a shorthand notation for the sum of outer products of the columns in factor matrices [21]:

$$
\mathcal{X} \approx [\![ \mathbf{A}_1, \cdots, \mathbf{A}_N ]\!],
$$

$$
\mathbf{X}_{(n)} \approx \mathbf{A}_n (\odot_{k \neq n} \mathbf{A}_k)^{\top}.
$$

Then, CP decomposition aims to find the factor matrices that minimize the estimation error $\mathcal{L}$ defined as follows:

$$
\mathcal{L}(\mathbf{A}_1, \cdots, \mathbf{A}_N) = \| \mathcal{X} - [\![ \mathbf{A}_1, \cdots, \mathbf{A}_N ]\!] \|^2 = \| \mathbf{X}_{(n)} - \mathbf{A}_n (\odot_{k \neq n} \mathbf{A}_k)^{\top} \|^2. \tag{6}
$$

CP alternating least squares (CP–ALS) has been extensively used for this optimization problem. The main idea of ALS is to divide the original problem into $N$ sub-problems, where each sub-problem corresponds to updating one factor matrix while keeping all the others fixed [20]:

$$
\mathbf{A}_n \leftarrow \arg\min_{\mathbf{A}_n} \| \mathbf{X}_{(n)} - \mathbf{A}_n (\odot_{k \neq n} \mathbf{A}_k)^{\top} \|^2
$$

$$
= \mathbf{X}_{(n)} [(\odot_{k \neq n} \mathbf{A}_k)^{\top}]^{\dagger} = \mathbf{X}_{(n)} (\odot_{k \neq n} \mathbf{A}_k)(\circledast_{k \neq n} \mathbf{A}_k^{\top} \mathbf{A}_k)^{\dagger}. \tag{7}
$$

## Online tensor decomposition

Of particular interest in the problem of tensor decomposition is an efficient online algorithm for time-evolving tensors. We think of a tensor as a set of "slices" given at each time step. Given an $N$-order time-evolving tensor $\boldsymbol{\mathcal{X}} \in \mathbb{R}^{I_1 \times \cdots \times I_N}$, we expand it as a form of $[\boldsymbol{\mathcal{X}}^{old}, \boldsymbol{\mathcal{X}}^{new}]^T$, where $\boldsymbol{\mathcal{X}}^{old} \in \mathbb{R}^{I_1^{old} \times \cdots \times I_N}$ is the previous tensor data and $\boldsymbol{\mathcal{X}}^{new} \in \mathbb{R}^{I_1^{new} \times \cdots \times I_N}$ is a new tensor slice for one time step. Then, the goal is to efficiently decompose the tensor $\boldsymbol{\mathcal{X}}$ given the previous decomposition result $\boldsymbol{\mathcal{X}}^{old} \approx [\![\tilde{\mathbf{A}}_1, \cdots, \tilde{\mathbf{A}}_N]\!]$ :

$$\boldsymbol{\mathcal{X}} = \begin{bmatrix} \boldsymbol{\mathcal{X}}^{old} \\ \boldsymbol{\mathcal{X}}^{new} \end{bmatrix} \approx \left[\!\!\left[ \begin{bmatrix} \mathbf{A}_1^{(0)} \\ \mathbf{A}_1^{(1)} \end{bmatrix}, \mathbf{A}_2, \cdots, \mathbf{A}_N \right]\!\!\right], \tag{8}$$

where $\mathbf{A}_1^{(0)} \in \mathbb{R}^{I_1^{old} \times R}$ and $\mathbf{A}_1^{(1)} \in \mathbb{R}^{I_1^{new} \times R}$. This is done by minimizing the estimation error $\mathcal{L}$ defined as follows:

$$\mathcal{L} = \frac{1}{2} \left\| \boldsymbol{\mathcal{X}}^{old} - [\![\mathbf{A}_1^{(0)}, \cdots, \mathbf{A}_N]\!] \right\|^2 + \frac{1}{2} \left\| \boldsymbol{\mathcal{X}}^{new} - [\![\mathbf{A}_1^{(1)}, \cdots, \mathbf{A}_N]\!] \right\|^2. \tag{9}$$

## Related works

Tensor stream decomposition is widely studied under CP decomposition [22, 23]. Existing works employ one of the following two ideas: they update (1) only the non-temporal factors with precomputed auxiliary matrices [14], or (2) whole factors considering prior decomposition results [15, 16]. We describe three main approaches (OnlineCP, SeekAndDestroy, and DTD) for dynamic tensor decomposition, and compare them with our proposed method.

## OnlineCP

OnlineCP [14] preserves the previous temporal factor to efficiently decompose new tensor slices. After updates of non-temporal factors and the partial temporal factor, it simply appends a part of the temporal factor matrix to the previous matrix. OnlineCP avoids duplicated computations such as Khatri-Rao and Hadamard products by introducing auxiliary matrices. It computes complementary matrices before ALS iteration and yields a new decomposition. Despite its low computational cost, the approach cannot achieve an accurate decomposition due to the lack of consideration on the change of themes in data (see Fig 2). Note that DAO-CP solves this problem by tracking local error norms of the tensor stream and detecting a change point of themes, which enables an accurate decomposition even when the data have an inconsistent temporal pattern.

## SeekAndDestroy

SeekAndDestroy [16] additionally uses rank estimation to discover latent concepts and detect concept drift in streaming tensors. The method estimates the rank of each incoming tensor slice, and updates the previous decomposition after alleviating concept drift. However, SeekAndDestroy requires extra computation due to the rank estimation for every time step, which causes a substantial loss of speed. Moreover, it consistently performs worse than OnlineCP when the initial rank of OnlineCP is fine-tuned. Note that our proposed method efficiently detects the change of theme in streaming data because it does not require estimating

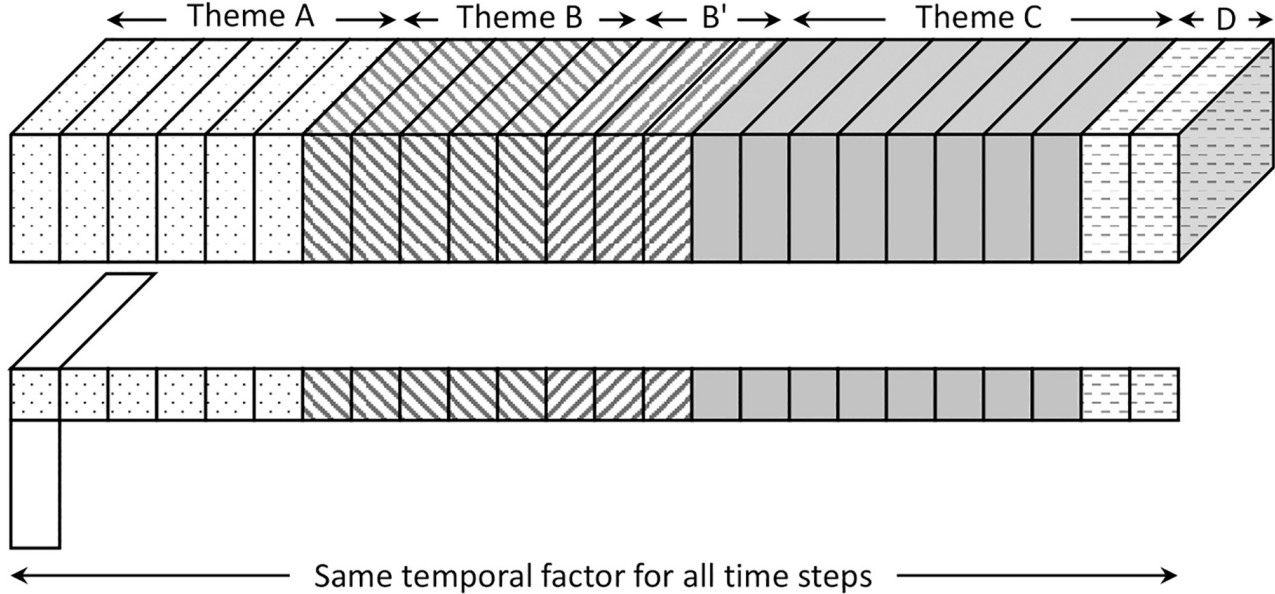

**Fig 2. Visualization of OnlineCP.** In this figure, the length of time factor (with horizontal axis) becomes larger for each time step. Contrary to static decomposition methods, OnlineCP reduces computational cost using approximation with an additional constraint: it updates only the non-temporal mode, reusing the same temporal factor for all time steps. However, this leads to a substantial loss of accuracy if an incoming tensor has a different theme compared to previous tensors (e.g., theme changes $A \rightarrow B$, $B' \rightarrow C$, or $C \rightarrow D$). Thus, OnlineCP cannot achieve an accurate decomposition due to the lack of consideration on the change of themes in data.

the actual rank numbers, but only tracks local error in order to rapidly capture the change points.

## DTD

DTD [15] was originally introduced as a part of MAST which is a low-rank tensor completion method to fill in the missing entries of the incomplete multi-aspect tensor stream. The method manages to reduce the time complexity by reusing the previous decomposition that approximates the tensor stacked until new slices come in. Specifically, for an $N$-th order tensor stream, DTD partitions the data into $2^N$ sub-tensors for each time step and uses binary tuples $(i_1, \cdots, i_N) \in \Theta = \{0, 1\}^N$ to denote the sub-tensors. Then, given that $[\![\tilde{\mathbf{A}}_1, \cdots, \tilde{\mathbf{A}}_N]\!]$ approximates $\boldsymbol{\mathcal{X}}^{(0,\cdots,0)} := \boldsymbol{\mathcal{X}}^{old}$, one can reformulate the estimation error $\mathcal{L}$ of online tensor decomposition as follows (see the notations from Preliminaries):

$$
\begin{aligned}
\mathcal{L} &= \mu \left\| [\![\tilde{\mathbf{A}}_1, \cdots, \tilde{\mathbf{A}}_N]\!] - [\![\mathbf{A}_1^{(0)}, \cdots, \mathbf{A}_N^{(0)}]\!] \right\|^2 + \mathcal{L}_0, \\
\mathcal{L}_0 &= \sum_{(i_1,\cdots,i_N) \in \Theta \setminus (0,\cdots,0)} \left\| \boldsymbol{\mathcal{X}}^{(i_1,\cdots,i_N)} - [\![\mathbf{A}_1^{(i_1)}, \cdots, \mathbf{A}_N^{(i_N)}]\!] \right\|^2,
\end{aligned}
\tag{10}
$$

where $\mu \in [0, 1]$ is the forgetting factor which alleviates the influence of the previous decomposition error. Although DTD is an efficient method, it suffers from poor accuracy when an incoming tensor has an entirely different pattern compared to previous tensors, as it still tries to reuse the prior decomposition result. Our proposed method addresses the problem by using

"re-decomposition" process and adapting quickly to sudden changes in data, and significantly increases the accuracy of decomposition.

## Proposed method

We propose DAO-CP, an accurate and efficient online algorithm for tensor stream decomposition.

### Overview

DAO-CP is a time and memory efficient algorithm for accurate online CP–ALS tensor decomposition which adapts to data changes. The challenge of decomposing time-evolving tensors is to improve accuracy without sacrificing speed and memory usage. Considering that the themes of data change over time, we propose detecting the change points of themes and using different strategies depending on the degree of change. The main challenges are as follows:

1. **Reduce computational cost**. How can we reduce the arithmetic cost for updating decomposition factors of tensor streams?

2. **Identify themes in data streams**. How can we capture the latent themes in tensor streams and detect the change points of them?

3. **Increase decomposition accuracy**. How can we exploit the detected change points of themes and increase the decomposition accuracy?

To address the above challenges, we propose the following approaches.

1. **Build an updatable framework for tensor stream**. We use *complementary matrices* and previous decomposition results recursively, where the complementary matrices are updated only when there is a change in non-temporal factors, thus reducing the redundant operations.

2. **Detect data changes by tracking error norms**. We continuously track the error norms of incoming data slices in the tensor stream, detecting a sudden accuracy drop based on *z-score* analysis, which we regard as a change point of themes.

3. **Re-decompose the tensor stream when a new theme is detected**. Once a sudden change in theme is detected, we choose whether to *refine* or *split* the tensor stream depending on the degree of changes. We also introduce *memory rate* to improve the refinement process. These techniques determine how much information from the previous decomposition should be retained, balancing the trade-off between accuracy and speed.

### Update rules for DAO-CP

Let $\boldsymbol{\mathcal{X}} = [\boldsymbol{\mathcal{X}}^{old}, \boldsymbol{\mathcal{X}}^{new}]^T$ be an $N$-order time-evolving tensor, where $\boldsymbol{\mathcal{X}}^{old} \in \mathbb{R}^{I_1^{old} \times \cdots \times I_N}$ is the previous tensor data and $\boldsymbol{\mathcal{X}}^{new} \in \mathbb{R}^{I_1^{new} \times \cdots \times I_N}$ is a new tensor slice; we assume that the first mode is the temporal mode. We design our update rules to efficiently decompose the tensor $\boldsymbol{\mathcal{X}} \approx [\![\mathbf{A}_1, \cdots, \mathbf{A}_N]\!]$, given the previous decomposition result $\boldsymbol{\mathcal{X}}^{old} \approx [\![\tilde{\mathbf{A}}_1, \cdots, \tilde{\mathbf{A}}_N]\!]$. We partition the temporal factor matrix $\mathbf{A}_1$ into old and new parts as $\mathbf{A}_1 = [\mathbf{A}_1^{(0)}, \mathbf{A}_1^{(1)}]^T$ using $\mathbf{A}_1^{(0)} \in \mathbb{R}^{I_1^{old} \times R}$ and $\mathbf{A}_1^{(1)} \in \mathbb{R}^{I_1^{new} \times R}$, where $R$ is the decomposition rank.

In order to consider the degree of change in themes, we introduce the *memory rate* $\rho \in [0.5, 1]$ which determines how much weight to assign to the decomposition of the previous tensor

data. We define the estimation error $\mathcal{L}$ as a restricted form of the one from DTD [15], where the non-temporal modes of the tensor stream are fixed:

$$\mathcal{L} = \frac{\rho}{2} \left\| \left[\!\left[ \tilde{\mathbf{A}}_1, \cdots, \tilde{\mathbf{A}}_N \right]\!\right] - \left[\!\left[ \mathbf{A}_1^{(0)}, \cdots, \mathbf{A}_N \right]\!\right] \right\|^2 + \frac{1}{2} \left\| \boldsymbol{\mathcal{X}}^{new} - \left[\!\left[ \mathbf{A}_1^{(1)}, \cdots, \mathbf{A}_N \right]\!\right] \right\|^2. \qquad (11)$$

The optimization of the estimation error $\mathcal{L}$ is based on CP-ALS [14, 20]. Note that we simplify the estimation error from DTD by setting the changes in non-temporal modes to zeros because there is a change only in the temporal mode for our problem. The update rules to minimize $\mathcal{L}$ in (11) for each factor matrix are derived as follows:

$$\mathbf{A}_1^{(0)} \leftarrow \tilde{\mathbf{A}}_1 (\circledast_{k\neq1} \tilde{\mathbf{A}}_k^\top \mathbf{A}_k)(\circledast_{k\neq1} \mathbf{A}_k^\top \mathbf{A}_k)^\dagger,$$

$$\mathbf{A}_1^{(1)} \leftarrow \boldsymbol{\mathcal{X}}_{(1)}^{new} (\odot_{k\neq1} \mathbf{A}_k)(\circledast_{k\neq1} \mathbf{A}_k^\top \mathbf{A}_k)^\dagger,$$

$$\mathbf{A}_{i\neq1} \leftarrow [\rho\tilde{\mathbf{A}}_i(\circledast_{k\neq1,i} \tilde{\mathbf{A}}_k^\top \mathbf{A}_k)\tilde{\mathbf{A}}_1^\top \mathbf{A}_1^{(0)} + \boldsymbol{\mathcal{X}}_{(i)}^{new}(\odot_{k\neq1,i} \mathbf{A}_k) \odot \mathbf{A}_1^{(1)}]$$
$$\times [(\circledast_{k\neq1,i} \mathbf{A}_k^\top \mathbf{A}_k)(\rho\mathbf{A}_1^{(0)\top} \mathbf{A}_1^{(0)} + \mathbf{A}_1^{(1)\top} \mathbf{A}_1^{(1)})]^\dagger.$$

Note that we also update the prior temporal factors to further increase the accuracy of decomposition. If the previous temporal factors are not updated, they harm the accuracy of method whenever there is a change of theme because they are optimized only for the previous theme of the data. However, it is computationally demanding to directly apply these recursive processes. To address the problem, we introduce two complementary matrices $\mathbf{G}$ and $\mathbf{H}$,

$$\mathbf{G} = \circledast_{k\neq1} \tilde{\mathbf{A}}_k^\top \mathbf{A}_k, \quad \mathbf{H} = \circledast_{k\neq1} \mathbf{A}_k^\top \mathbf{A}_k, \qquad (12)$$

where $\mathbf{G}$ and $\mathbf{H}$ are updated only when there is a change in non-temporal factors, thus reducing the redundant computations. This leads to the following modified update rules:

$$\mathbf{A}_1^{(0)} \leftarrow \tilde{\mathbf{A}}_1 \mathbf{G}\mathbf{H}^\dagger, \qquad (13)$$

$$\mathbf{A}_1^{(1)} \leftarrow \boldsymbol{\mathcal{X}}_{(1)}^{new}(\odot_{k\neq1} \mathbf{A}_k)\mathbf{H}^\dagger, \qquad (14)$$

$$\mathbf{A}_{i\neq1} \leftarrow [\rho\tilde{\mathbf{A}}_i(\mathbf{G} \oslash \tilde{\mathbf{A}}_i^\top \mathbf{A}_i)\circledast\tilde{\mathbf{A}}_1^\top \mathbf{A}_1^{(0)} + \boldsymbol{\mathcal{X}}_{(i)}^{new}(\odot_{k\neq1,i} \mathbf{A}_k) \odot \mathbf{A}_1^{(1)}]$$
$$\times [(\mathbf{H} \oslash \mathbf{A}_i^\top \mathbf{A}_i)\circledast(\rho\mathbf{A}_1^{(0)\top} \mathbf{A}_1^{(0)} + \mathbf{A}_1^{(1)\top} \mathbf{A}_1^{(1)})]^\dagger. \qquad (15)$$

The overall update process is outlined in Algorithm 1.

**Algorithm 1**: DAO-CP Alternating Least Square (DAO-CP-ALS)

```
Input: Factors from old tensors  ⟦Ã₁,···,Ã_N⟧ , new tensor slice 𝒳^new,
memory rate ρ, and number of ALS iterations n_iter
Output: Updated factors  ⟦A₁, ···, A_N⟧
1 Initialize complementary matrices G and H by (12)
2 for n ← 1 to n_iter do
3   Update A₁^(1) using (14)      // latter part of temporal factor A₁
4   for A_{i≠1} ∈ non-temporal factors do
5     Update A_i, G and H using (15)
6   Update A₁^(0) using (13)      // former part of temporal factor A₁
```

## Change points detection with local error norm

The key of our proposed method is to detect change points of themes in tensor streams and thereby adapt quickly to abrupt changes in data. To do this, we continuously track the decomposition error of tensor streams and detect a sudden accuracy drop which we regard as a change point of themes. Such an accuracy drop is captured by measuring *local error norm* $\mathcal{E}_{local}$ for the new tensor slice and its decomposition result:

$$\mathcal{E}_{local} := \left\| \boldsymbol{\mathcal{X}}^{new} - [\![ \mathbf{A}_1^{(1)}, \cdots, \mathbf{A}_N ]\!] \right\|^2. \tag{16}$$

We assume that $\mathcal{E}_{local}$ follows a normal distribution $\mathcal{E}_{local} \sim \mathcal{N}(\mu, \sigma^2)$ and keep track of its mean $\mu(\mathcal{E}_{local})$ and variance $\sigma^2(\mathcal{E}_{local})$ to detect outliers. Note that we should update the mean and variance in an online manner. This is achievable by using Welford's algorithm [24, 25], which provides accurate estimates of mean and variance without the necessity of keeping the entire data. Moreover, the method requires only one pass of given data in order to compute their sample mean and variance. Using Welford's algorithm, we detect outliers in the current local error norm by *z-score* analysis of the following criterion:

$$\text{z-score} = \frac{\mathcal{E}_{local} - \mu(\mathcal{E}_{local})}{\sigma(\mathcal{E}_{local})} > L, \tag{17}$$

where $L$ is a threshold of anomaly. Changing the value of $L$, one can fine-tune the criterion on whether a new tensor slice is similar to the previous tensors or not.

## Re-decomposition process

Once a sudden change of theme is detected by z-score analysis, DAO-CP exploits this information to increase the decomposition accuracy. As a new tensor slice is stacked for each time step, DAO-CP updates the factor matrices following the optimization scheme described in Algorithm 1. It then computes the z-score in the local error norm distribution and performs "re-decomposition" depending on the score. Tracking the distribution of local error norm and setting the z-score criteria enable DAO-CP to automatically choose the best strategy between *split* and *refinement* processes depending on the degree of changes. Fig 3 illustrates the intuition of the two processes, and Table 2 shows the criterion for each process.

**Split process.** What if an incoming tensor slice has an entirely different theme compared to previous tensors? In this case, reusing the prior results of decomposition will cause a substantial loss of accuracy. To address the problem, we design a split process which divides the streaming tensor into separate tensors of different themes, using a threshold $L_s$. Despite extra costs of space and time due to re-initialization, the split process enables DAO-CP to successfully avoid the unexpected accuracy drop (lines 8-13 in Algorithm 2).

**Refinement process.** The refinement process is used to update the decomposition result when there is only a modest difference in the theme from the previous tensor. We use the hyperparameter $L_r$ to fine-tune the refinement criterion on whether an incoming tensor slice is similar to the previous tensors or not. We use the memory rate $1 - \rho$ because we need to focus more on the new slice (note that $1 - \rho \leqslant \rho$ since $\rho \in [0.5, 1]$). The ALS operation also takes the z-score and is performed extra more times accordingly, because a higher z-score implies that there is a more abrupt change in data. As a result, these techniques determine how much information from the previous decomposition should be retained, balancing the trade-off between accuracy and running time (lines 14-16 in Algorithm 2). The full computation of DAO-CP is outlined in Algorithm 2.

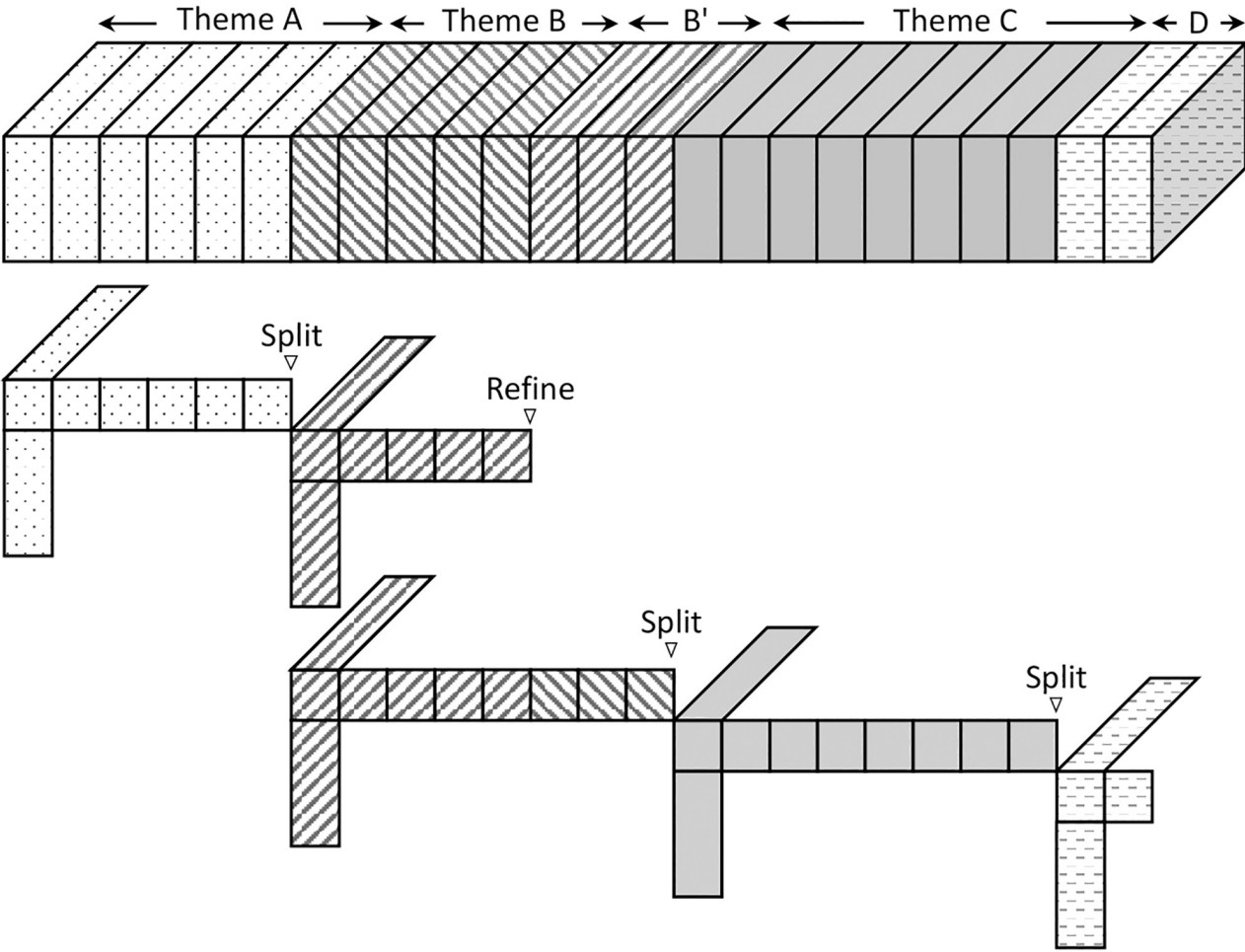

**Fig 3. Visualization of split and refinement processes of DAO-CP.** When the z-score exceeds the split threshold $L_s$ (e.g., change from theme $A$ to $B$), DAO-CP re-initializes the new tensor slice using the static CP decomposition. When the z-score is between $L_r$ and $L_s$ (e.g, change from theme $B$ to $B'$), the refinement process determines how much information from the previous factors should be retained. Consequently, DAO-CP provides both fast and accurate decomposition for tensor streams even with inconsistent temporal patterns.

**Algorithm 2**: Data-Adaptive Online CP Decomposition (DAO-CP)

```
Input: Tensor stream 𝒳_stream, memory rate ρ, and number of ALS iterations
       n_iter
Output: Decomposition factor set 𝒮 = { ⟦A₁,···,A_N⟧ }
1 𝒳_new ← new slice from 𝒳_stream
2 Initialize ⟦Ã₁,···,Ã_N⟧ using CP decomposition of 𝒳_new
```

**Table 2. Execution criteria for split and refinement processes.** $L_s$ is the threshold of splitting and initializing the decomposition, and $L_r$ is the threshold of refining the previous decomposition. By changing the two hyperparameters, we can fine-tune the re-decomposition process to balance the trade-off between accuracy and speed.

| Process | Criterion |
|---|---|
| Split | $z > L_s$ |
| Refinement | $L_r < z \leqslant L_s$ |
| - | $z \leqslant L_r$ |

```
 3 Calculate error norm 𝓔_local between 𝓧_new and ⟦Ã_1,⋯,Ã_N⟧
 4 Initialize Welford with 𝓔_local
 5 for 𝓧_new from 𝓧_stream do
 6    ⟦A_1, ⋯, A_N⟧ ← DAO-CP-ALS ( ⟦Ã_1,⋯,Ã_N⟧ , 𝓧_new, ρ, n_iter)
 7    Calculate error norm 𝓔_local between 𝓧_new and ⟦A_1, ⋯, A_N⟧
    /* Split Process */
 8    if Welford z-score (𝓔_local) > L_s then
 9       Store the previous factors to 𝓢
10       Initialize ⟦Ã_1,⋯,Ã_N⟧ using CP decomposition of 𝓧_new
11       Calculate error norm 𝓔_local between 𝓧_new and ⟦Ã_1,⋯,Ã_N⟧
12       Initialize Welford with 𝓔_local
13       continue
    /* Refinement Process */
14    if L_s ⩾ Welford z − score (𝓔_local) > L_r then
15       ⟦A_1, ⋯, A_N⟧ ← DAO-CP-ALS
( ⟦Ã_1,⋯,Ã_N⟧ , 𝓧_new, 1 − ρ, (1 + z − score) · n_iter)
16       Calculate error norm 𝓔_local between 𝓧_new and ⟦A_1, ⋯, A_N⟧
17    Update Welford with 𝓔_local
18    ⟦Ã_1,⋯,Ã_N⟧ ← ⟦A_1,⋯,A_N⟧
19 Store the previous factors to 𝓢
```

## Theoretical analysis

We analyze the computational complexity of DAO-CP. The following symbols are used for the analysis: $N$ (dimensionality), $R$ (rank), $I_1^{new}$ (time length of the new data slice), $I_1^{old}$ (time length of the formerly stacked data), $I_{i\neq1}$ (mode length of the non-temporal $i$-th mode), and $n_{iter}$ (number of ALS iterations).

Table 3 summarizes the comparison of DAO-CP to existing tensor decomposition methods. We find that DAO-CP has the lowest arithmetic cost among the methods except OnlineCP. Note that even though OnlineCP has lower complexity than our proposed method, it suffers from poor accuracy due to the lack of consideration on temporal change of data, which limits its usage in practice (see Fig 4).

**Lemma 1**. *The time complexity of initializing the complementary matrices* **G** *and* **H** *by* (12) *is* $O(R^2 \sum_{k\neq1} I_k)$.

*Proof.* Because the operands $\tilde{\mathbf{A}}_k^\top \mathbf{A}_k$ and $\mathbf{A}_k^\top \mathbf{A}_k$ in Hadamard operations are $R \times R$ matrices, it takes $O(R^2 \cdot I_k)$ for multiplication of $\tilde{\mathbf{A}}_k^\top \in \mathbb{R}^{R \times I_k}$ and $\mathbf{A}_k \in \mathbb{R}^{I_k \times R}$. Thus, the total arithmetic complexity of computing the matrices **G** and **H** is given by $O(R^2 \cdot \sum_{k\neq1} I_k)$.

**Lemma 2**. *The time complexity of updating the factor matrix* $\mathbf{A}_1^{(0)}$ *by* (13) *is* $O(R^2 I_1^{old} + R^3)$.

*Proof.* Finding the pseudo-inverse matrix $\mathbf{H}^\dagger$ of **H** takes $O(R^3)$, and the matrix multiplication of $\tilde{\mathbf{A}}_1 \in \mathbb{R}^{I_1^{old} \times R}$ and $\mathbf{G}, \mathbf{H}^\dagger \in \mathbb{R}^{R \times R}$ requires $O(R^2 \cdot I_1^{old})$ operations.

**Lemma 3**. *The time complexity of updating the factor matrix* $\mathbf{A}_1^{(1)}$ *by* (14) *is* $O(R I_1^{new} \prod_{k\neq1} I_k + R^2 I_1^{new} + R^3)$.

**Table 3. Comparison of existing tensor decomposition methods (Full–CP refers to the static CP decomposition method).**

|  | Computational Complexity | Online | Updatable | Adaptive |
|---|---|:---:|:---:|:---:|
| Full–CP [26] | $O(NRI_1^{old}\prod_{i\neq1}I_i + NRI_1^{new}\prod_{i\neq1}I_i)$ |  | ✓ |  |
| OnlineCP [14] | $O(NRI_1^{new}\prod_{i\neq1}I_i)$ | ✓ |  |  |
| DTD [15] | $O(NRI_1^{new}\prod_{i\neq1}I_i + NR^2\sum_i I_i + NR^3)$ | ✓ | ✓ |  |
| DAO-CP | $O(NRI_1^{new}\prod_{i\neq1}I_i + R^2\sum_i I_i + NR^3)$ | ✓ | ✓ | ✓ |

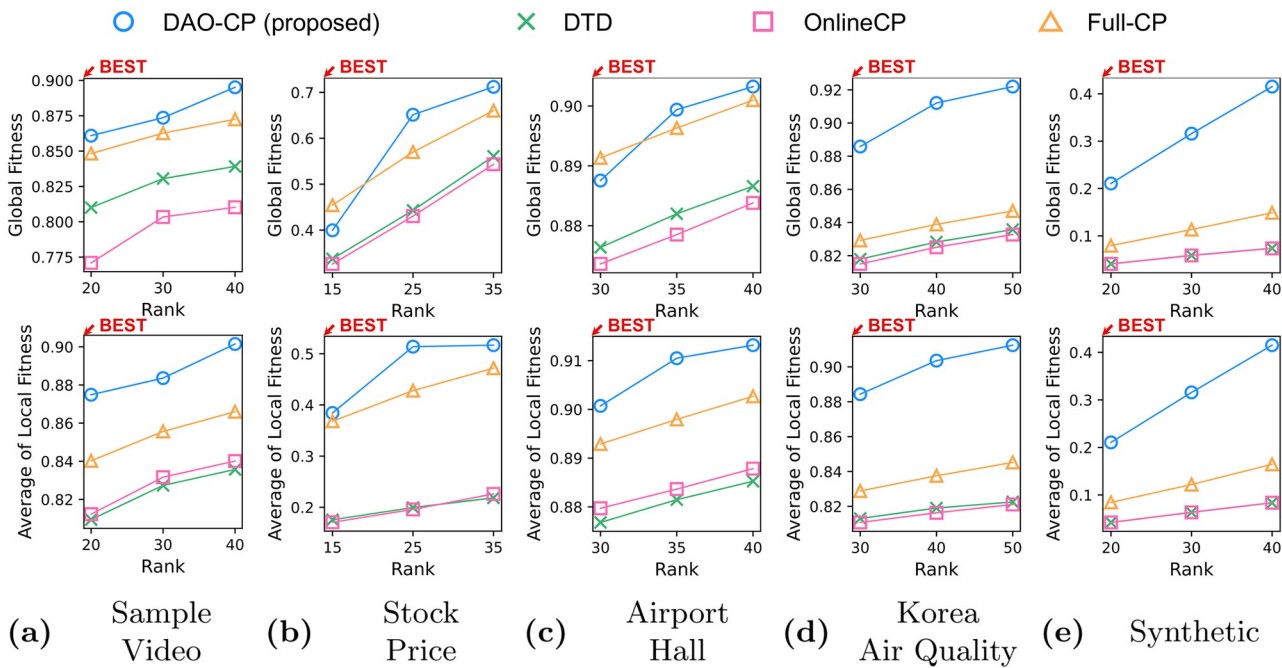

**Fig 4. Reconstruction errors: Global (upper) and local (lower) fitness.** Since Full–CP is not an online method, we evaluate its fitness whenever a new slice is added. Detecting the change points of theme, DAO-CP successfully increases the accuracy of decomposition, which is even higher than that of Full–CP.

*Proof.* The Khatri-Rao product $(\bigodot_{k \neq 1} \mathbf{A}_k) \in \mathbb{R}^{\prod_{k \neq 1} I_k \times R}$ consumes $O(R \cdot \prod_{k \neq 1} I_k)$. We compute the matrix multiplication in the order of appearance, and its time complexity is $O(R \cdot I_1^{new} \cdot \prod_{k \neq 1} I_k + R^2 \cdot I_1^{new})$. This, combined with the complexity $O(R^3)$ of finding $\mathbf{H}^\dagger$, yields the desired result.

**Lemma 4.** *The time complexity of updating* $\mathbf{A}_{i \neq 1}$, $\mathbf{G}$ *and* $\mathbf{H}$ *by* (15) *is* $O(RI_1^{new} \prod_{k \neq 1} I_k + R^2 I_i + R^3)$.

*Proof.* Computing $\mathbf{A}_i^\top \mathbf{A}_i$, $\mathbf{A}_1^{(0)\top} \mathbf{A}_1^{(0)}$ and $\mathbf{A}_1^{(1)\top} \mathbf{A}_1^{(1)}$ require $O(R^2 \cdot I_i)$, $O(R^2 \cdot I_1^{old})$ and $O(R^2 \cdot I_1^{new})$ time costs, respectively. Constructing the pseudo-inverse matrix costs $O(R^3)$ since its size is $R \times R$. For the first part before the pseudo-inverse, the first term is calculated in $O(R^2 \cdot I_i + R^2 \cdot I_{old})$ time, and the second term is computed similarly to Lemma 3 with $O(R \cdot I_1^{new} \prod_{k \neq 1} I_k)$ complexity. Combining those results, we conclude the proof.

**Theorem 5** (Complexity of DAO-CP-ALS). *The time complexity of* DAO–CP–ALS *(Algorithm 1) is* $O(n_{iter}(NRI_1^{new} \prod_{i \neq 1} I_i + R^2 \sum_i I_i + NR^3))$.

*Proof.* The computational cost of updating all the non-temporal factor matrices $\{\mathbf{A}_{i \neq 1}\}$ is $\sum_{i \neq 1} O(RI_1^{new} \prod_{k \neq 1} I_k + R^2 I_i + R^3)$ by Lemma 4, which can be written as $O(NRI_1^{new} \prod_{i \neq 1} I_i + R^2 \sum_{i \neq 1} I_i + NR^3)$. Combining the complexities of updating $\{\mathbf{A}_i\}$ by Lemmas 2 and 3 gives the following arithmetic cost for a single ALS iteration: $O(NRI_1^{new} \prod_{i \neq 1} I_i + R^2 \sum_i I_i + NR^3)$. Thus, including the initialization cost from Lemma 1, we obtain the desired result.

## Experiments

In this section, we experimentally evaluate DAO-CP to answer the following questions.

- **Q1. Reconstruction error**. How accurately does DAO-CP decompose real-world tensor streams compared to existing methods?

- **Q2. Time cost**. How much does DAO-CP improve the speed of tensor stream decomposition compared to existing online methods?

- **Q3. Effect of thresholds $L_r$ and $L_s$**. How do the different choices of $L_r$ and $L_s$ for re-decomposition criteria affect the performance of DAO-CP?

- **Q4. Refinement and split processes**. How does each of the re-decomposition processes affect the performance of DAO-CP on real-world datasets?

In the following, we describe the experimental settings and answer the questions with the experimental results.

## Experimental settings

All the experiments are conducted in a workstation with a single CPU (Intel(R) Xeon(R) CPU E5-2630 v4 @ 2.20GHz).

**Datasets.** We use four real-world tensor streams and a synthetic tensor stream summarized in Table 4, where the first mode of each tensor corresponds to the temporal mode. We construct a tensor stream by splitting an original tensor into slices along the time mode (first mode); e.g., we make 41 slices from Sample Video dataset, where each slice is of dimension (5, 240, 320, 3).

- Sample Video dataset is a series of animation frames with RGB values. For this dataset, we expect that changes of theme occur when an object starts moving or a scene changes.

- Stock Price includes data on 140 stocks listed on the Korea Stock Price Index 200 (KOSPI 200) from Jan 2, 2008 to June 30, 2020, where each stock contains five features: adjusted opening price, closing price, highest price, lowest price, and trading volume. Changes of theme may occur when real-world events affect the rise or fall of stock markets.

- Airport Hall is a video recorded in an airport, initially used to verify OLSTEC [17, 27]. We expect that a sudden change of theme occurs when a crowd of people surges toward the airport during flight departure or arrival time.

- Korea Air Quality dataset consists of daily air pollutant levels for various locations in Seoul, South Korea from Sep 1, 2018 to Sep 31, 2019. The themes may continuously change depending on weather environment.

- Synthetic is made of concatenated tensors, which is the summation $T_{main} + T_{theme} + T_{noise}$ of three tensors $\{T_{main}, T_{theme}, T_{noise}\}$, each referring to a $\{\mathcal{N}(0, 100), \mathcal{N}(0, 10), \mathcal{N}(0, 1)\}$

**Table 4. Summary of datasets.**

| Datasets | Order | Dimensions | Slice Length | Rank | $L_s$ | $L_r$ |
|---|---|---|---|---|---|---|
| Sample Video[1] | 4 | (205, 240, 320, 3) | 5 | 30 | 6.0 | 2.0 |
| Stock Price[1] | 3 | (3089, 140, 5) | 3 | 20 | 6.0 | 5.0 |
| Airport Hall[1] | 3 | (200, 144, 176) | 10 | 20 | 0.5 | 0.1 |
| Korea Air Quality[1] | 3 | (9479, 323, 6) | 100 | 20 | 2.0 | 1.3 |
| Synthetic[1] | 4 | (1000, 10, 20, 30) | 10 | 30 | 1.2 | 1.1 |

[1] https://github.com/snudatalab/DAO-CP/

normally distributed randomized tensor, respectively, of size (1000, 10, 20, 30). Note that we simulate the changes of theme using the tensor $T_{theme}$.

**Competitors.** We compare DAO-CP with existing dynamic tensor decomposition methods including OnlineCP [14] and DTD [15], as well as with the static CP decomposition method, Full–CP [20]. All the methods are implemented in Python3 using the TensorLy library.

**Parameters.** The parameters $L_s$ and $L_r$ of DAO-CP are set to the values listed in Table 4. The section "Effect of Thresholds $L_s$ and $L_r$" is an exception, where we vary the two values to investigate the effect of different thresholds. We set the memory rate as $\rho = 0.8$ for all the experiments.

**Evaluation measure.** To evaluate our proposed method, we use local and global error norms $\mathcal{E}_{local}$ and $\mathcal{E}_{global}$, as well as the corresponding "fitness" scores $\mathcal{F}_{local}$ and $\mathcal{F}_{global}$, which are defined as follows:

$$\mathcal{E}_{local} = \| \boldsymbol{\mathcal{X}}^{new} - [\![ \mathbf{A}_1^{(1)}, \cdots, \mathbf{A}_N ]\!] \|^2, \quad \mathcal{E}_{global} = \| \boldsymbol{\mathcal{X}} - [\![ \mathbf{A}_1, \cdots, \mathbf{A}_N ]\!] \|^2,$$

$$\mathcal{F}_{local} = 1 - \frac{\mathcal{E}_{local}}{\| \boldsymbol{\mathcal{X}}^{new} \|}, \quad \mathcal{F}_{global} = 1 - \frac{\mathcal{E}_{global}}{\| \boldsymbol{\mathcal{X}} \|}.$$

$\mathcal{F}_{local}$ denotes the fitness for an incoming data slice at each time step, while $\mathcal{F}_{global}$ is the fitness for whole tensors. They are the normalized versions of error norms with respect to data size, designed to compare the decomposition accuracy for multiple datasets with different sizes.

**Running time.** We evaluate the speed of each method in terms of local running time, which is the elapsed time for decomposing the current data slice. Because Full–CP is not an online algorithm, we assume that it decomposes the entire tensor whenever a new data slice comes in.

## Reconstruction error

We compare DAO-CP to its competitors in terms of fitness, varying the decomposition rank in Fig 4. The average of local fitness is the mean of $\mathcal{F}_{local}$ that is computed at every time step. Note that DAO-CP shows higher fitness than the existing methods in most cases, regardless of ranks.

## Running time

DAO-CP allows an accurate tensor factorization by exploiting the characteristic of data and detecting change points. However, this results in a slightly longer running time due to the re-decomposition process. Fig 5 shows the running times of DAO-CP and other methods for various ranks. Note that DAO-CP has moderate running times between the static and dynamic decomposition methods, showing promising speeds comparable to the other dynamic algorithms (DTD and OnlineCP) and significantly faster than the static method (Full–CP).

## Effect of thresholds $L_r$ and $L_s$

We change the values of $L_s$ and $L_r$ to investigate the effect of split and refinement processes. Table 5 shows the results, where the number of refinement or split points changes as $L_r$ and $L_s$ vary. Note that both the processes lead to more accurate decomposition with extra time costs, and among them the split process has bigger trade-offs because it requires re-initialization. As

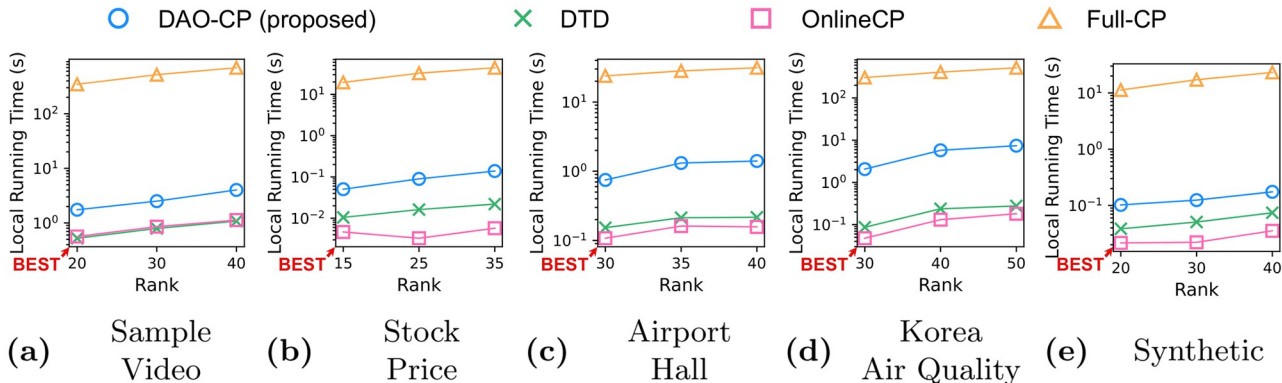

**Fig 5. Time cost: Average of local running time.** Since Full–CP is not an online method, we evaluate its fitness whenever a new slice is added. Note that DAO-CP results in a promising speed comparable to DTD and OnlineCP with much more accurate decomposition, and significantly faster than Full–CP.

a result, the more parts the tensor is split into (smaller $L_s$), the more accurate decomposition DAO-CP yields with extra costs of time. More refinement processes (smaller $L_r$) also have a similar effect, although the trade-offs are relatively small. In contrast to the split process, it requires more memory to store intermediate data such as auxiliary matrices **G** and **H**. In a practical standpoint, these observations are very useful because one can benefit from the hyperparameter tuning when there is a particular importance in one of accuracy, speed, or memory usage.

### Refinement and split processes

Recall that the split process is used to start a new decomposition when an entirely different theme is detected, while the refinement process is used when there is only a modest difference from the previous decomposition. Fig 1 validates the importance of these intuitions, showing that DAO-CP results in remarkable performance for the video datasets with different scenes and object movements.

**Table 5. Effect of thresholds $L_r$ and $L_s$.** The memory usage means the summation of byte allocation to store intermediate data to calculate next decomposition results (e.g. auxiliary matrices **G** and **H**). We use Korea Air Quality dataset with rank 20, and change $L_r$ and $L_s$ to investigate the effect of refinement and split processes. Note that the lower the thresholds is set, the more frequently the re-decomposition processes are executed. Thus, one can benefit from this observation when there is a particular importance in one of accuracy, speed, or memory usage depending on target tasks.

| Process | $L_r$ | $L_s$ | # of executions | Running time | Memory usage | Fitness |
|---|---|---|---|---|---|---|
| None | - | - | 0 | 12.72 sec | 1,649 KB | 80.50% |
| Refinement | 2.2 | - | 6 | 12.48 sec | 1,698 KB | 80.62% |
| | 2.0 | - | 8 | 12.56 sec | 1,698 KB | 80.65% |
| | 1.8 | - | 12 | 12.96 sec | 1,954 KB | 80.67% |
| | 1.6 | - | 14 | 13.25 sec | 2,082 KB | 80.70% |
| | 1.4 | - | 15 | 13.39 sec | 2,082 KB | 80.72% |
| Split | - | 1.8 | 5 | 24.05 sec | 949 KB | 82.17% |
| | - | 1.6 | 36 | 125.24 sec | 405 KB | 85.27% |
| | - | 1.4 | 45 | 147.56 sec | 405 KB | 85.72% |
| | - | 1.2 | 57 | 184.39 sec | 405 KB | 86.34% |
| | - | 1.0 | 65 | 211.42 sec | 277 KB | 87.56% |

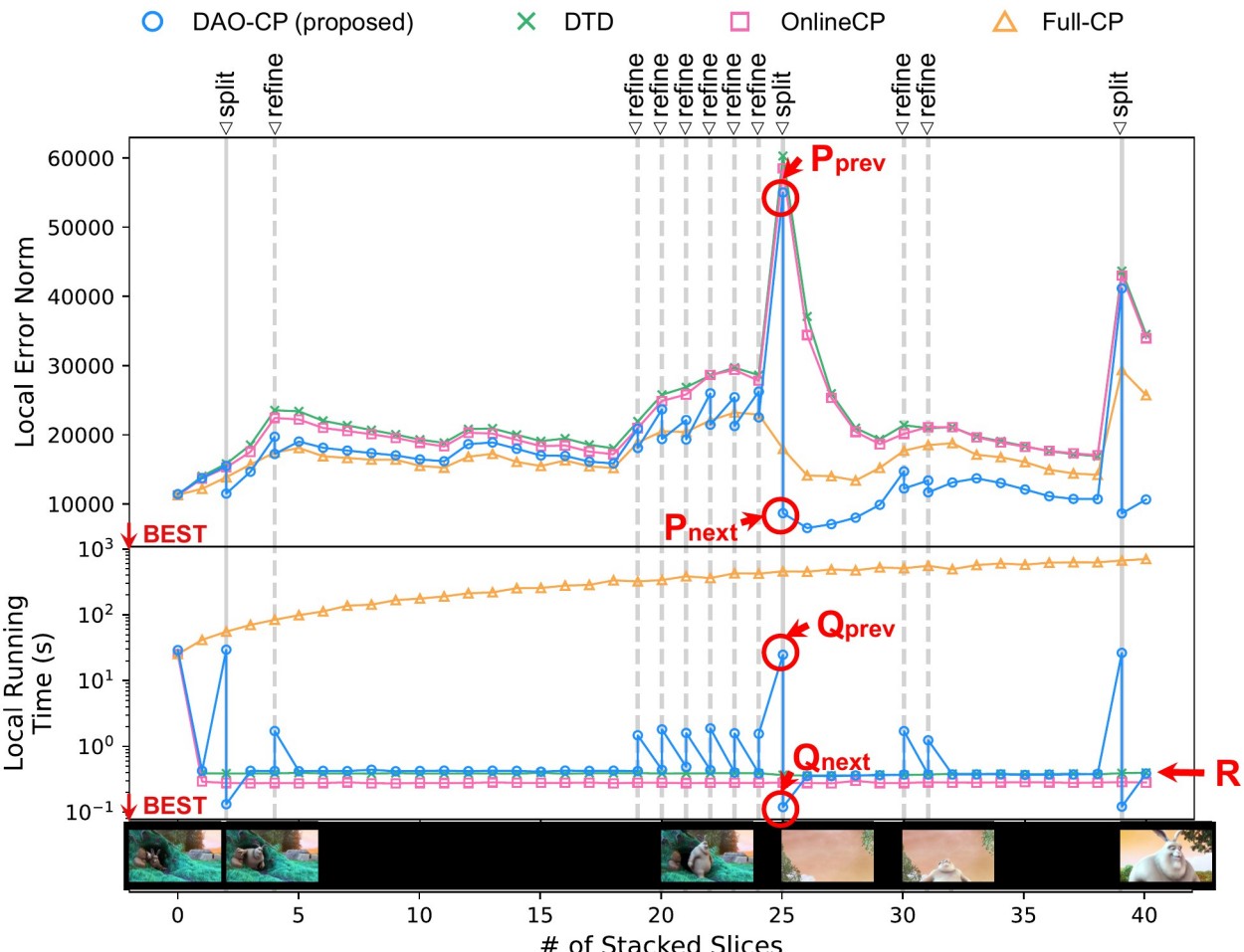

**Fig 6. Refinement and split processes: Effects of split (solid line) and refinement (dashed line) processes in terms of local error norm (upper) and running time (lower).** Each re-decomposition process (at split point) significantly reduces the local error norm with only a modest sacrifice of running time (e.g., vertical line connecting $P_{prev}$, $P_{next}$, $Q_{prev}$, and $Q_{next}$). Note that DAO-CP runs slower than the other dynamic methods (OnlineCP and DTD) only when one of split or refinement processes is performed to increase the accuracy (horizontal line R: average running time of competitor methods).

To further investigate the effects of split and refinement processes, we consider the following question: for each time step (or data slice), how does the re-decomposition process affect the running time and local error norm? With Sample Video dataset, we compare the running time and local error norm of DAO-CP to its competitors in Fig 6. We observe that both split and refinement processes significantly reduces the local error norm with only a modest sacrifice of running time.

## Conclusions

In this paper, we propose DAO-CP, an efficient algorithm for decomposing time-evolving tensors. DAO-CP automatically detects a change point of theme in tensor streams and decides whether to re-decompose the tensors or not. Experimental results show that the proposed DAO-CP outperforms the current state-of-the-art methods on both synthetic and real-world datasets. We also investigate the effect of hyperparameters of our proposed method and demonstrate the advantages of trading-off between accuracy, speed, and memory usage. Future

works include extending our method for simultaneously decomposing many related time-evolving tensors.

## Author Contributions

**Conceptualization:** Sangjun Son, Yong-chan Park, U. Kang.

**Data curation:** Sangjun Son, Yong-chan Park, Minyong Cho.

**Formal analysis:** Sangjun Son, Yong-chan Park.

**Investigation:** Sangjun Son, Yong-chan Park, Minyong Cho.

**Methodology:** Sangjun Son.

**Supervision:** U. Kang.

**Validation:** Yong-chan Park.

**Writing – original draft:** Sangjun Son, Yong-chan Park.

**Writing – review & editing:** Yong-chan Park, U. Kang.

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
