## [Decision Letter · Decision Letter 0]

11 Feb 2022

PONE-D-21-39294DAO-CP: Data-Adaptive Online CP Decomposition for Tensor StreamPLOS ONE

Dear Dr. Kang,

Thank you for submitting your manuscript to PLOS ONE. After careful consideration, we feel that it has merit but does not fully meet PLOS ONE’s publication criteria as it currently stands. Therefore, we invite you to submit a revised version of the manuscript that addresses the points raised during the review process.

We look forward to receiving your revised manuscript.

Kind Regards,

Ning Cai, Ph.D.

Academic Editor

PLOS ONE

Journal Requirements:

“This work was supported by the National Research Foundation of Korea(NRF) funded by MSIT(2019R1A2C2004990).

The Institute of Engineering Research and ICT at Seoul National University provided research facilities for this work.”

Reviewers' comments:

Reviewer's Responses to Questions

**Comments to the Author**

1. Is the manuscript technically sound, and do the data support the conclusions?

Reviewer #1: Partly

Reviewer #2: Yes

2. Has the statistical analysis been performed appropriately and rigorously? 

Reviewer #1: N/A

Reviewer #2: N/A

3. Have the authors made all data underlying the findings in their manuscript fully available?

Reviewer #1: Yes

Reviewer #2: Yes

4. Is the manuscript presented in an intelligible fashion and written in standard English?

Reviewer #1: Yes

Reviewer #2: Yes

5. Review Comments to the Author

Reviewer #1: In this paper, the authors propose DAO–CP, an accurate and efficient online CP decomposition method, which adapts to data changes. DAO–CP tracks local error norms of the tensor streams, detecting a change point of the error norms. It then chooses the best strategy depending on the degree of changes to balance the trade-off between speed and accuracy.

I have the following comments for this paper:

1. The format of this article is not standard. If you use word template, please align left and right.

2. The writting is bad. I am not so sure about your contributions after reading the paper. your contribution is just the rules to detect the check points with local error norm? You use alternating minimization for DAO-CP.

3. Due to your method is data adaptive, it is better to give an depict of the data, which may affect the computation of local error norm?

4. It may be better to add some comparisons with some existing online methods.

Reviewer #2: Overall the paper is fairly easy to follow, and is consistent in notation, terminology, and theme as many other works in this area. Since the streaming decomposition method is basically DTD, the primary technical contribution of the paper is the logic for adapting the decomposition based on error thresholds when the current factor matrices do not well represent the new tensor data. Their logic for the adaptation process is reasonable, but does include some hyperparameters that would require tuning for each new problem. I think that contribution is significant enough to merit publication, but I do have several comments outlined in the attached review.

6. PLOS authors have the option to publish the peer review history of their article (what does this mean?). If published, this will include your full peer review and any attached files.

Reviewer #1: No

Reviewer #2: No

---

## [Author Response · Author response to Decision Letter 0]

22 Mar 2022

We would like to thank the reviewers for their high quality reviews and constructive comments. We hope our revision has successfully addressed all your concerns.

Below, we typed the content in 'rebuttal_letter.pdf' for just in case.

Reviewer 1.

• (R1-1) The format of this article is not standard. If you use word template, please align left and right.

– (A1-1) We corrected the format of the article according to reviewer’s comments.

• (R1-2) The writting is bad. I am not so sure about your contributions after reading the paper. your contribution is just the rules to detect the check points with local error norm? You use alternating minimization for DAO-CP.

– (A1-2) We revised the paper to clearly highlight the contributions of our work (lines 29-35). Our main contributions are summarized as follows: 

(1) We employ z-score analysis to rapidly detect the change points of streaming tensors. 

(2) We introduce re-decomposition process to balance the trade-off between accuracy and speed. 

(3) We use complementary matrices and simplify the objective function from DTD in order to reduce the redundant computations in CP-ALS optimization.

• (R1-3) Due to your method is data adaptive, it is better to give an depict of the data, which may affect the computation of local error norm?

– (A1-3) We further included description of the data in Experiments section (lines 308-326). For example, Fig. 6 illustrates how our proposed method adapts to the Sample Video dataset; DAO-CP achieves the state-of-the-art accuracy when an object starts moving or a scene changes in the data.

• (R1-4) It may be better to add some comparisons with some existing online methods.

– (A1-4) We included another recent paper called “Identifying and Alleviating Concept Drift in Streaming Tensor Decomposition (R. Pasricha et al., 2019),” and compared it with our proposed method in Related Works section (lines 116-125).

Reviewer 2.

• (R2-1) The approach does not appear to be appropriate for infinite data streams because by my reading of Eq. 12, all prior time steps are included in first term defining the objective function for the optimization problem solved each time step. Typically, authors use an exponential down-weighting of older and older time steps to effectively truncate the prior temporal data that must be stored and manipulated. Furthermore, this can be seen in Lemma 2, showing a cost proportional to I1old, which would grow linearly in time. It seems to me this would be straightforward improvement, so I wonder why the authors chose not to do this.

• (R2-2) It is also unusual that the authors update the factor matrix rows corresponding to the prior time steps (see first line after Eq. 12). Typically, authors use a 2-stage process where they update the temporal mode keeping the non-temporal mode fixed (which means the rows of the factor matrix corresponding to old time steps do not change), and then update the nontemporal modes using the newly updated temporal factor. This saves computational cost, so again, I wonder why the authors chose not to do this. 

– (A2-1 and A2-2) We choose to consider all prior time steps and update the whole temporal factors to further increase the accuracy of decomposition. If the previous temporal factors are not updated, they harm the accuracy of method whenever there is a change of theme because they are optimized only for the previous theme of the data. Thus, even though our proposed method requires a time cost growing linearly, it significantly increases the decomposition accuracy (Fig. 4), and shows promising speeds comparable to state-of-the-art methods for all the datasets (Fig. 5). We included additional explanation to justify our method in Proposed Method section (lines 190-193).

• (R2-3) The authors should make it clear their approach does not adapt the rank of the CP decomposition, nor add or remove factors. They should discuss how their method compares to the concept drift approach in SeekAndDestroy (Parischa, Gujral, Papelaxakis, 2019). 

– (A2-3) Unlike SeekAndDestroy, DAO-CP does not require estimating the actual rank numbers, but only tracks the difference of them using local error norms. This technique enables DAO-CP to rapidly capture the change points of theme without adding or removing factors. We included additional comparisons between DAO-CP and SeekAndDestroy in Related Works section (lines 116-125).

• (R2-4) The computational results compare to running the full CP decomposition each time step using the entire streamed tensor to that point, which isn’t necessary. A more fair comparison would be just running full CP at the end of the stream, since running it at intermediate time steps does not provide any new information.

– (A2-4) We compared the computation results between DAO-CP and full CP for each time step because we aim to answer how well DAO-CP performs for time-evolving tensors. In Fig. 6, we find that DAO-CP outperforms full CP in terms of local error norm whenever refinement or split process is executed. If we run full CP only at the end of the stream, we cannot compare the local error norm between DAO-CP and full CP at intermediate time steps.

---

## [Decision Letter · Decision Letter 1]

4 Apr 2022

DAO-CP: Data-Adaptive Online CP Decomposition for Tensor Stream

PONE-D-21-39294R1

Dear Dr. Kang,

We’re pleased to inform you that your manuscript has been judged scientifically suitable for publication and will be formally accepted for publication once it meets all outstanding technical requirements.

Kind regards,

Ning Cai, Ph.D.

Academic Editor

PLOS ONE

Additional Editor Comments (optional):

Reviewers' comments:

Reviewer's Responses to Questions

**Comments to the Author**

1. If the authors have adequately addressed your comments raised in a previous round of review and you feel that this manuscript is now acceptable for publication, you may indicate that here to bypass the “Comments to the Author” section, enter your conflict of interest statement in the “Confidential to Editor” section, and submit your "Accept" recommendation.

Reviewer #3: All comments have been addressed

Reviewer #4: All comments have been addressed

2. Is the manuscript technically sound, and do the data support the conclusions?

Reviewer #3: Yes

Reviewer #4: Yes

3. Has the statistical analysis been performed appropriately and rigorously? 

Reviewer #3: Yes

Reviewer #4: Yes

4. Have the authors made all data underlying the findings in their manuscript fully available?

Reviewer #3: Yes

Reviewer #4: Yes

5. Is the manuscript presented in an intelligible fashion and written in standard English?

Reviewer #3: Yes

Reviewer #4: Yes

6. Review Comments to the Author

Reviewer #3: (No Response)

Reviewer #4: In this paper, authors proposed online CP decomposition method which adapts to data changes. The method can better balance the trade-off of the tensor streams between speed and accuracy. The results of this paper is interesting and important.

7. PLOS authors have the option to publish the peer review history of their article (what does this mean?). If published, this will include your full peer review and any attached files.

Reviewer #3: No

Reviewer #4: No

---

## [Editor Report · Acceptance letter]

6 Apr 2022

PONE-D-21-39294R1 

DAO-CP: Data-Adaptive Online CP Decomposition for Tensor Stream 

Dear Dr. Kang:

I'm pleased to inform you that your manuscript has been deemed suitable for publication in PLOS ONE. Congratulations! Your manuscript is now with our production department. 

Kind regards, 

on behalf of

Dr. Ning Cai 

Academic Editor

PLOS ONE